# Variation in Compressive Strength of Concrete aross Thickness of Placed Layer

**DOI:** 10.3390/ma12132162

**Published:** 2019-07-05

**Authors:** Jarosław Michałek

**Affiliations:** Faculty of Civil Engineering, Wrocław University of Science and Technology, 50-370 Wrocław, Poland; jaroslaw.michalek@pwr.edu.pl; Tel.: +48-71-320-2264

**Keywords:** concrete slabs and floorings, horizontal casting, compressive strength, ultrasonic tests

## Abstract

Is the variation in the compressive strength of concrete across the thickness of horizontally cast elements negligibly small or rather needs to be taken into account at the design stage? There are conflicting answers to this question. In order to determine if the compressive strength of concrete varies across the thickness of horizontally cast elements, ultrasonic tests and destructive tests were carried out on core samples taken from a 350 mm thick slab made of class C25/30 concrete. Special point-contact probes were used to measure the time taken for the longitudinal ultrasonic wave to pass through the tested sample. The correlation between the velocity of the longitudinal ultrasonic wave and the compressive strength of the concrete in the slab was determined. The structure of the concrete across the thickness of the slab was evaluated using GIMP 2.10.4. It was found that the destructively determined compressive strength varied only slightly (by 3%) across the thickness of the placed layer of concrete. Whereas the averaged ultrasonically determined strength of the concrete in the same samples does not vary across the thickness of the analyzed slab. Therefore, it was concluded that the slight increase in concrete compressive strength with depth below the top surface is a natural thing and need not be taken into account in the assessment of the strength of concrete in the structure.

## 1. Introduction

The view that the compressive strength of concrete varies across the thickness of horizontally cast elements (concrete slabs, floorings, etc.) is seldom expressed in the literature on the subject. Opinions as to the significance of this variation are widely divided, as the following survey of literature indicates.

The research published by Stawiski [1,2,3] provided the direct incentive for this study of the distribution of concrete compressive strength along the height of horizontally cast elements. On the basis of ultrasonic tests of core samples taken from concrete, Stawiski found the compressive strength of the concrete to be lower in the top zone than in the bottom zone by as much as 40–50% [1,2,3]. The variation in compressive strength along the height of the cross section was approximately linear. The fall in ultrasonic wave velocity at the sample’s top surface is ascribed by Stawiski [1] to the surface weakening effect connected with concrete consolidation resulting in the segregation of concrete components. The main factor responsible for the decrease in concrete compressive strength is considered to be porosity, which very strongly affects ultrasonic wave velocity. Also the inadequate curing of fresh concrete, damage to the structure of concrete caused by corrosion, and mechanical damage to the top surface of the concrete which can arise in the course of the service life of the element are also possible factors.

Stawiski [3] proposed to introduce (besides the grade of concrete) strength gradient ∇fc into the evaluation of concrete in horizontally cast elements (e.g., floor toppings). On the basis of his research [3] Stawiski pointed out that in, e.g., an approximately 15 cm thick element the strength gradient of the concrete at the depth of 10 cm from the bottom amounted to 0.7 MPa/cm, whereas in the layers situated closer to the top surface it varied markedly (−3.0, −4.5, −8.0 MPa/cm). Therefore, Stawiski calls for [3] defining allowable variations in concrete compressive strength, e.g., ∇f_c_ ≤ 1.0 MPa/cm. The increase of 1.0 MPa/cm in the strength of concrete in the lower situated layers relative to the top layer suggested by Stawiski [3] seems to be very large.

On the basis of ultrasonic tests of the compressive strength across the thickness of samples taken from cut out pieces of 40, 45 and 60 mm thick floorings made of cement mortars, Hoła, Sadowski and Hoła A. [4] found the strength was not identical and varied across the thickness. The lowest strength was in the top zone, the highest in the bottom zone, while in the middle zone, it was close to the destructively determined compressive strength. In the considered case, the strength gradient of the mortar across the thickness of the flooring amounted to 6–7 MPa/cm.

Petersons in [5] found the compressive strength of the lower situated layers to be higher than that of the top layer, but only by 10–20%. No further increase in concrete strength was observed in the layers situated below 300 mm. The difference in compressive strength between the top surface and the bottom surface in slabs was ascribed to the inadequate curing of the concrete [5].

In monograph [6], Dąbrowski, Stachurski and Zieliński found that the deeper situated layers of concrete had higher strength than the surface layer. Below 80 cm, this increase in strength stabilized at the level of approximately 10%. In the authors’ opinion [6], this is due to the well-known property of concrete—it reaches higher strength when hardening under a moderate pressure—and that is why this phenomenon does not occur in samples of low height.

Yuan, Ragab, Hill and Cook [7] found that the compressive strength of concrete along the height of the placed layer did not vary significantly. Suprenant [8] found that the compressive strength of concrete in slabs varied minimally, and only in a small upper part of the element. The most marked variation in concrete strength along element height has been observed in walls and beams. This is mainly due to the greater static pressure exerted by the concrete situated above.

Neville [9] found that the slight increase in concrete compressive strength below the top surface was a natural thing, but need not be taken into account. When testing a reinforced concrete wall and beam by means of the ultrasonic method, Watanabe, Hishikawa, Kamae and Namiki [10] found the compressive strength of concrete in samples taken from the lower part of the element was slightly higher than in samples taken from its upper part. They treated this as a natural thing which did not need to be taken into account.

Neville [11] ascribed the variation in the compressive strength of concrete along the height of the sample to the presence of retained water, occurring during concrete bleeding.

According to standard [12], the compressive strength of concrete in a structure can be lower in the top layer than in the bottom layer by as much as 25%. Concrete characterized by lower compressive strength usually occurs to a depth of 300 mm or to 20% of the height of the cross section, depending on which of the values is lower. According to standard [13], the range of variation in the compressive strength of concrete in a structure can differ between the particular portions of the structure. The variation is random and often forced (by, e.g., the relative density, the degree of compaction, the curing conditions, etc.).

Therefore, the questions arises: Is the variation in the compressive strength of concrete across the thickness of horizontally cast elements negligibly small or rather needs to be taken into account at the design stage?

## 2. Description of Author’s Investigations

### 2.1. Ultrasonic Tests of Concrete

In order to verify the phenomenon of concrete compressive strength variation across the thickness of horizontally cast elements, samples with diameter d = 100 mm height h = 350 mm (Table 1), taken from a specially cast slab made of concrete C25/30 (the grade of the concrete was determined using concrete cubes cast when casting the slab) were subjected to ultrasonic tests. CEMII/BS-32.5 cement (270 kg/m^3^), fly ash additive (60 kg/m^3^), plasticizer (2.43 kg/m^3^), water (170 kg/m^3^) and 1879 kg/m^3^ of natural aggregate (sand 0/2 mm −40%, gravel 2/8 mm −26%, gravel 8/16 mm −34%) were used in the concrete mix for the slab construction. The latter had been compacted by means of an immersion vibrator and cured for 28 in the laboratory conditions defined in standard [14]. The slab had been exposed to variable weather conditions for two years. Samples (01−06 in Figure 1) were drilled out of the slab perpendicularly to its top surface. For reference purposes specimens (07−12) were drilled out of the slab parallel with its top surface. Prior to the tests, the actual dimensions of the samples and their weight were determined (Table 1).

The measuring points spaced at every 1 cm (Figure 1 and Figure 2) were marked on the sides of the core samples. Velocity C_L_ of ultrasonic wave passage through concrete was measured in two perpendicular directions. No concrete/probe coupling material was used. The probes were set perpendicularly to the tested side surface of the sample (Figure 2). The distributions of velocity C_L_ of longitudinal ultrasonic wave passage through the concrete were obtained from the tests (Table 2 and Table 3, and Figure 3 and Figure 4).

A Unipan Materials Tester Type 543 with point-contact exponential probes [15] and a frequency of 40 kHz (Figure 2) was used to measure the time taken for the longitudinal ultrasonic wave to pass through the tested sample. Prior to the tests, the instrument had been calibrated to determine the time taken for the ultrasonic wave to pass through the probes alone (t_0_ = 36.6 μs). The details of the operation of exponential heads with point-to-point contact with the examined surface are described in detail in the paper [15].

Table 2 and Table 3 and Figure 3 and Figure 4 showed that the distributions of velocity C_L_ of the longitudinal ultrasonic wave along the height of the core samples with d ≅ 100 mm were quite constant (except for the top layers). The mean velocity C_L_ of ultrasonic wave propagation in the samples (01–06) taken perpendicularly to the top surface of the slab was C_L1_ = 3.54 km/s, a standard deviation s_CL1_ = 0.13 km/s and a variation coefficient ν_CL1_ = 3.67%. In the case of the samples (07–12) taken parallel with the top of the slab, the following were obtained: C_L2_ = 3.45 km/s, s_CL2_ = 0.072 km/s and ν_CL2_ = 2.09%. On the basis of the obtained velocities C_L_ of the longitudinal ultrasonic wave in the range: C_L_ = 3.5–4.5 km/s, it can be assessed [16] whether the quality of the concrete as good, whereas the velocities in the range of 3.0–3.5 km/s indicated dubious quality.

The observed slight fluctuations of velocity C_L_ of the longitudinal ultrasonic wave in the inner layers of the concrete are due to local concrete defects (e.g., air voids) or local strengthening with larger aggregate grains. Lower velocities C_L_ of the longitudinal ultrasonic wave were registered in the samples (01–06) taken perpendicularly to the top surface of the slab (Figure 3 and Figure 5). In the samples (07–12) taken parallel with the top of the slab, a decline in velocity CL of the longitudinal ultrasonic wave was observed only at the edge constituting the side edge of the slab (Figure 4 and Figure 6).

Initially, it was though that the falls in velocity C_L_ of the longitudinal ultrasonic wave were due to the sample end effect. Ultimately, it was decided that this phenomenon in samples 01–06 was caused from the top by bleeding [17] and concrete sedimentation, and from the bottom by the improper vibration of the concrete by means of the immersion vibrator (the vibrator was not fully immersed in the freshly placed concrete). In samples 07–12, the fall in velocity C_L_ of the longitudinal ultrasonic wave can be caused by the wall effect [9,10]. Figure 5 and Figure 6 show the structure of the concrete along the height of the samples taken perpendicularly to and parallel with the top surface of the slab. The image of the structure of the samples in Figure 5 and Figure 6 was prepared in GIMP 2.10.4 using the filter: LCHH(ab) component with a contrast of 50%. In Figure 5 the altered structure of the concrete is visible in the sample’s upper part (an approximately 30–40 mm thick layer) and lower part (an approximately 30–50 mm thick layer). In these places, reduced velocities of the longitudinal wave velocity were observed. In Figure 6, the altered structure of the concrete can be seen in an approximately 20–80 mm thick layer located at the side wall of the slab. Also in this layer, falls in the velocity of the longitudinal ultrasonic wave were recorded. 

### 2.2. Ultrasonic Tests of Concrete

For further investigations the core samples (Table 1) were cut into smaller samples (Figure 7) with height h = 100 mm (h/d = 1). In the terminal samples (e.g., 01/1—the top of the sample, 01/3—the bottom of the sample) the actual end faces were left unchanged (or were slightly trimmed to make them level). The middle sample (e.g., 01/2) was cut to the required size of 100 mm. Then, the end faces were prepared by grinding for compressive strength tests (Figure 8).

The compressive strength tests were carried out in conformance with standard [13] in the ZD100 strength testing machine (Figure 9a) satisfying the requirements of standard [18]. All the samples showed the same type of failure (Figure 9b). The parameters of the samples and the test results are presented in Table 2.

The concrete strength values (Table 4) yielded by the tests carried out on the100 mm high samples were used to evaluate the class of the concrete and the variation of strength along the core sample height (h = 350 mm) and were correlated with the results obtained using the ultrasonic method. According to standard [19], the result of concrete compressive strength tests carried out on cylindrical specimens with diameter d = 100 mm and height h = 100 mm, cut out of a structure directly corresponded to the strength of concrete determined on 150 × 150 × 150 mm standard cubes (f_ck,is_ = f_ck,is,cube_).

Standard [19] states that due to drilling, which undoubtedly can slightly damage the core’s material, the strengths of core samples determined in-situ are usually lower than the strengths of the standard samples. For this reason, it is allowed to use a correction factor of 0.85, understood as a ratio of the in-situ characteristic compressive strength to the characteristic compressive strength determined on the standard samples. As a result, the concrete compressive strength values coming directly from strength tests are increased.

The mean compressive strength of the concrete, determined on the 18 samples taken perpendicularly to the top surface of the slab, amounted to f_m(18),is_ = 31.45 MPa (the minimum value f_is,lowest_ = 30.58 MPa). The mean standard deviation amounted to s = 0.49 MPa. The coefficient k_1_ = 1.48 was assumed. The characteristic compressive strength of the concrete in the structure (f_ck,is_) was determined on the basis of standard [13], from the condition: f_ck,is_ = min(f_m(18),is_ − k_1_ × s; f_is,lowest_ + 4) = (30.72 MPa, 34.58 MPa). Thus the characteristic cube compressive strength of the concrete determined on samples taken perpendicularly to the top surface of the slab amounted to f_ck,is_ = f_ck,is,cube_ = 30.72 MPa.

The mean compressive strength of the concrete determined on the 18 samples taken parallel with the top surface of the slab amounted to f_m(18),is_ = 31.11 MPa (the minimum value f_is,lowest_ = 29.60 MPa). The mean standard deviation amounted to s = 0.81 MPa. The characteristic cube compressive strength of the concrete, determined on the samples taken parallel with the top surface of the slab, amounted to f_ck,is_ = (29.91 MPa, 33.60 MPa) = 29.91 MPa. The cube strength determined on the samples taken parallel to the top surface of the slab was 3% lower than the strength determined on the samples taken perpendicularly to the top surface of the slab. This confirms the observation that the strength of core samples drilled out horizontally is lower (on average by 8% [12,13]) than that of core samples drilled out vertically.

On the basis of the obtained f_ck,is,cube_ values, the actual strength class of the concrete in the structure is estimated to be f_ck,is,cube_ = 30.72 MPa and 29.91 MPa, respectively. When the correction factor of 0.85 is applied, this gives the concrete strength class respectively f_ck,is,cube_ = 36.1 MPa and 35.2 MPa, which corresponds to at least concrete class C25/30.

### 2.3. Scaling of Correlation Curve

On the basis of the measurements, the mean longitudinal ultrasonic wave passage velocities C_L_ were correlated with the mean compressive concrete strengths f_is_. When calculating the mean longitudinal ultrasonic wave passage velocity C_L_, the values from the areas of disturbances near the ends of the samples were rejected. The correlation curve was scaled according to the procedure described in standard [13] (version 2). In accordance with [16], a hypothetical base regression curve for ordinary concrete, i.e., f_CL,b_ = 2.39C_L_^2^ − 7.06C_L_ + 4.2 for C_L_ = 2.4–5.0 km/s, was adopted. Then, the differences δf between experimental compressive strength f_is_ and the strength obtained from base curve f_CL_, as well as the mean value δf_m(n)_ of the differences and standard deviation s were determined. The shift of the base correlation curve was calculated from the relation Δf = δf_m(n)_ − k_1_·s for coefficient k_1_ = 1.48 [13]. Ultimately, the corrected correlation curve f_CL_ = f_CL,b_ + Δf has the form f_CL_ = 2.39C_L_^2^ − 7.06C_L_ + 25.09. The obtained curve only slightly differs from the curves determined separately for samples 01 ÷ 06 and 07–12.

The obtained correlation was evaluated using two accuracy characteristics, i.e., the correlation coefficient η > 0.75 and the mean square relative deviation ν_k_ ≤ 12 ≤ %. The correlation coefficient amounted to:
η = [0.25 × ∑(f_CL,I_ − f_CL(36),ν_)^2^]^1/2^ ÷ [0.25 × ∑(f_is_ − f_m(36),is_)^2^]^1/2^η = [0.25 × 15.34]^1/2^ ÷ [0.25 × 16.33]^1/2^ = 0.97 > 0.75(1)
and the mean square relative deviation to:
ν_k_ = 100 × {[1/(*n* − 1)] × ∑[(f_CL,i_ − f_is_)/f_CL,i_]^2^}^1/2^ν_k_ = 100 × [(1/35)] × 0.12555]^1/2^ = 5.99% < 12%.(2)

Thus, it can be said that a good correlation between the mean longitudinal ultrasonic wave passage velocities C_L_ and the mean concrete compressive strengths f_is_ was obtained.

The class of the concrete in the structure was determined on the basis of the concrete compressive strength values obtained from the correlation curve f_CL_ = 2.39 × C_L_^2^ − 7.06C_L_ + 25.09 for the mean longitudinal ultrasonic wave passage velocities C_L_. The mean compressive strength of the concrete determined using the ultrasonic method amounted to f_CL(36),is_ = 29.80 MPa (minimal f_CL,is,lowest_ = 28.80 MPa) and the mean standard deviation to s = 0.66 MPa. Hence, the characteristic compressive strength of the concrete amounted to f_ck,is_ = f_ck,is,cube_ ≤ (29,80 – 1.48 × 0.66, 28.80 + 4) = (28.82 MPa, 32.80 MPa) = 28.82 MPa. When the correction factor of 0.85 was taken into account, concrete class C25/30 was obtained. The concrete class determined on the basis of the compressive strength values obtained from the correlation curve confirmed the destructively determined class of the concrete.

## 3. Analysis of Test Results

With the corrected correlation curve: f_CL_ = 2.39·C_L_^2^ − 7.06·C_L_ + 25.09, it was possible to trace the variation of the compressive strength of the concrete along the height of the analyzed samples. Figure 10 and Figure 11 show the variations of concrete compressive strength along the height of the core samples respectively, and perpendicular to (Figure 10) and parallel with (Figure 11) the top surface of the slab. Further, the results of the destructive tests and the averaged results of the ultrasonic tests for the particular samples with a height h = 100 mm are included in the diagrams.

An analysis of the concrete compressive strength values for the particular samples 01–06 (Figure 10) taken perpendicularly to the top plane of the slab indeed showed a slight increase (by 3%) in this strength in the sample’s lower part relative to its upper part. A similar phenomenon (also an increase by approximately 2.6%) was observed for samples 07–12 (Figure 11) taken parallel with the top plane of the slab. However, it should be noted that the compressive strength values were strongly averaged for the samples and included the effect of various factors connected with the destructive test itself.

The averaged compressive strength results obtained from the ultrasonic measurements showed (Figure 10 and Figure 11), however, that there was no increase in the compressive strength of the concrete along the height of the sample. This applies to the samples taken both perpendicularly to and parallel with the top plane of the slab.

The ultrasonic tests indicate that the variation in the compressive strength of concrete along the height of the sample is minimal and random. It can even be considered as negligible. The obtained results do not corroborate Stawiski’s theses [1,2,3], but confirm the results reported in [7,8,9,10].

The decreases in the compressive strength of the concrete occurring at the ends of the samples taken perpendicularly to the top plane of the slab (samples 01–06) and at the edge constituting the side edge of the slab for the samples taken parallel with the top plane of the slab (samples 07–12) were found to be interesting.

## 4. Conclusions

The following conclusions emerge from the investigations of the compressive strength of concrete carried out on core samples taken perpendicularly to and parallel with the top surface of the approximately 35 cm thick element, using different testing methods (the ultrasonic method and the destructive method):

The concrete compressive strength destructively determined along the height of the placed layer of concrete changed slightly (by 3%—samples 01–06) with a depth below the top surface of the element. The averaged concrete strength determined on the basis of the ultrasonic tests of the same samples did not vary across the thickness of the analyzed slab.
The obtained compressive strength increments across the thickness of the placed layer of concrete do not corroborate Stawiski’s theses [1,2,3], but confirm the results reported in, [7,8,9]. Therefore, there can be agreement with Neville’s statement [10] that the slight increase in concrete compressive strength with depth below the top surface is a natural thing and need not to be taken into account in the evaluation of the strength of concrete in the structure.The concrete compressive strength determined on core samples only slightly depends on the depth of where the sample came from (provided the ingredients of the concrete do not segregate as it is being placed and compacted).The use of the ultrasonic method for testing concrete with point-contact exponential probes showed the variation in concrete strength along the height of the core sample could be quite accurately evaluated and areas of lower quality concrete could be indicated. This was mainly from the thick layer (approximately 30–40 mm) extending from the top edge and the thick layer (approximately 30–50 mm) extending from the bottom edge of the samples 01–06 taken perpendicularly to the upper plane of the element. Further, from the thick layer (approximately 20–80 mm) extending from the edge constituting the side plane of the slab for the samples taken parallel with the top plane of the element (samples 07–12). The decline in the strength of the concrete in the upper part of samples 01–06 is caused by the bleeding of water from the concrete mixture (the bleeding phenomenon [17]) and the sedimentation of the latter. While in the lower part of the samples, it is due to the improper vibration of the placed layer of concrete mixture. The decrease in concrete strength at the side edge of the slab in the case of samples 07–12 can be due to the wall effect [9,10].From the point of view of the assessment of the concrete structure, supplementary tests on the slab in the future should be carried out using ultrasonic tomography [20,21].The ultrasonic method of testing concrete by means of point-contact exponential probes enables the accurate assessment of the quality of concrete (the segregation of concrete components, porosity, density, strength, etc.) along the height of a core sample drilled out perpendicularly to the placed layer.

## Figures and Tables

**Figure 1 materials-12-02162-f001:**
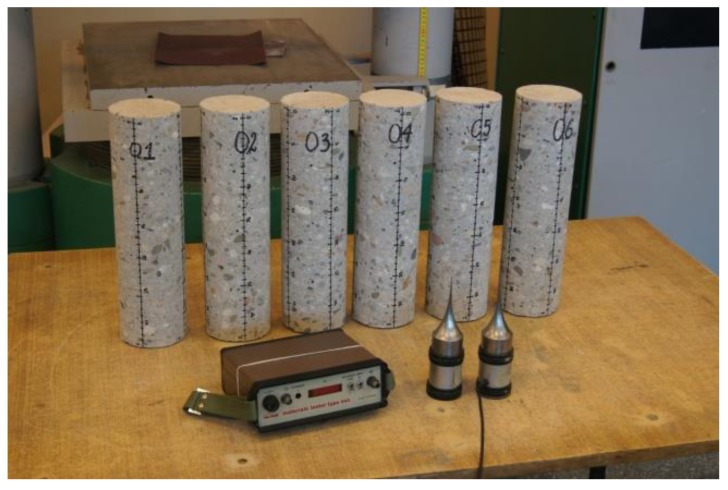
The samples to be tested using ultrasonic device.

**Figure 2 materials-12-02162-f002:**
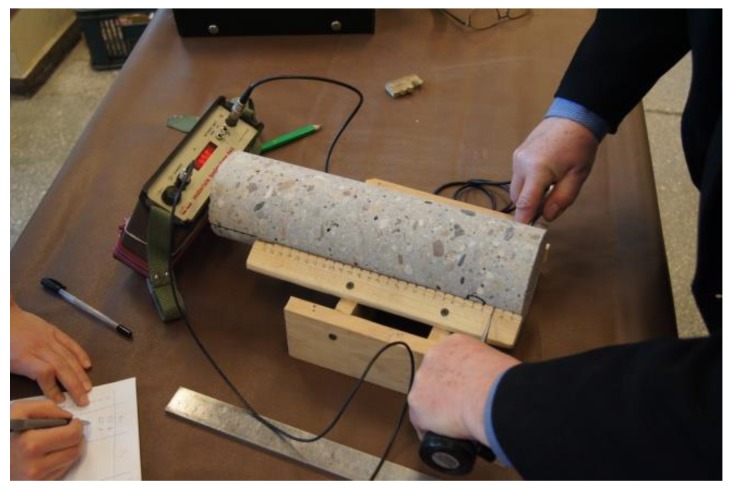
The measurement of velocity C_L_ of longitudinal ultrasonic wave.

**Figure 3 materials-12-02162-f003:**
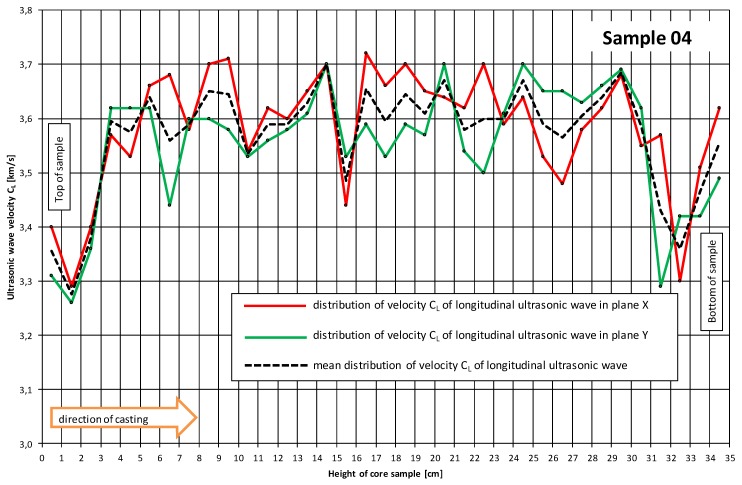
Exemplary distribution of velocity C_L_ of longitudinal ultrasonic wave along the height of sample 04 taken perpendicularly to the top surface of the slab.

**Figure 4 materials-12-02162-f004:**
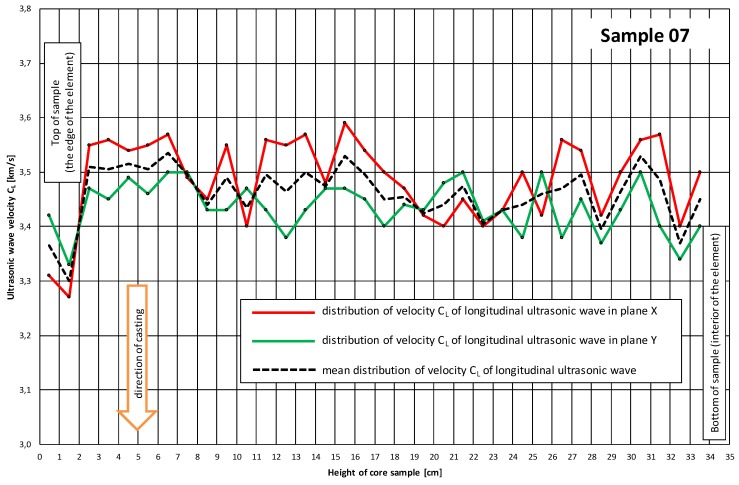
Exemplary distribution of velocity C_L_ of longitudinal ultrasonic wave along height of sample 07 taken parallel with the top surface of the slab.

**Figure 5 materials-12-02162-f005:**
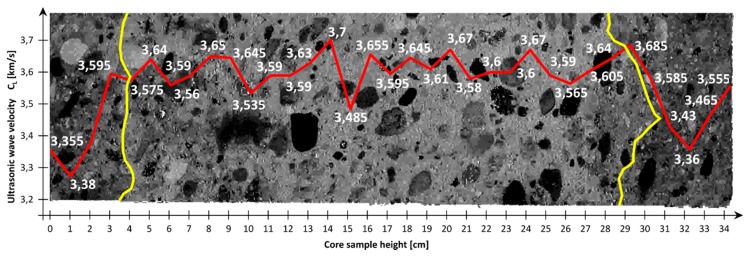
The structure of concrete of the sample taken perpendicularly to the top surface of the slab (slab top surface on left). The image also shows distribution of the mean velocity C_L_ of longitudinal ultrasonic wave (red line) and disturbance zone at the top and bottom surface of the slab (yellow line).

**Figure 6 materials-12-02162-f006:**
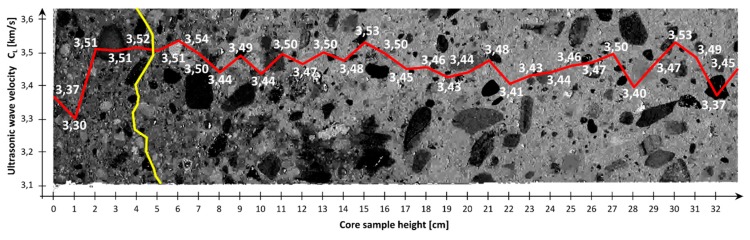
The structure of concrete of the sample taken parallel with the top surface of the slab (slab top surface on left). The image also shows distribution of the mean velocity C_L_ of longitudinal ultrasonic wave (red line) and zone of disturbance at the lateral surface of the slab (yellow line).

**Figure 7 materials-12-02162-f007:**
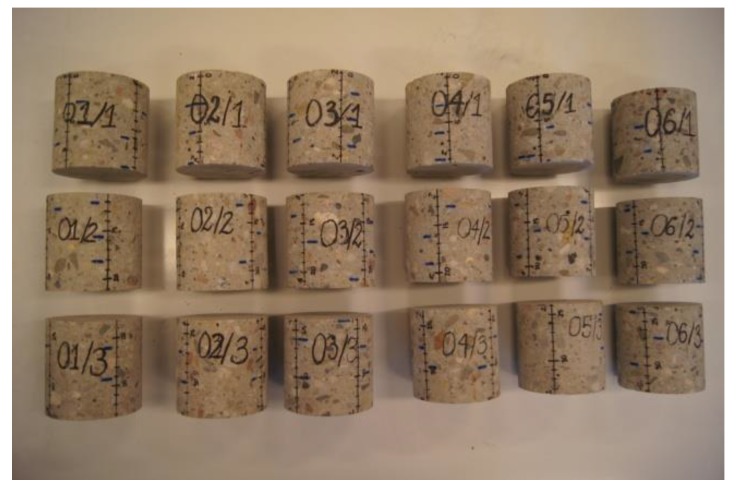
Core samples with height/diameter ratio h/d = 1, obtained from samples 01–06.

**Figure 8 materials-12-02162-f008:**
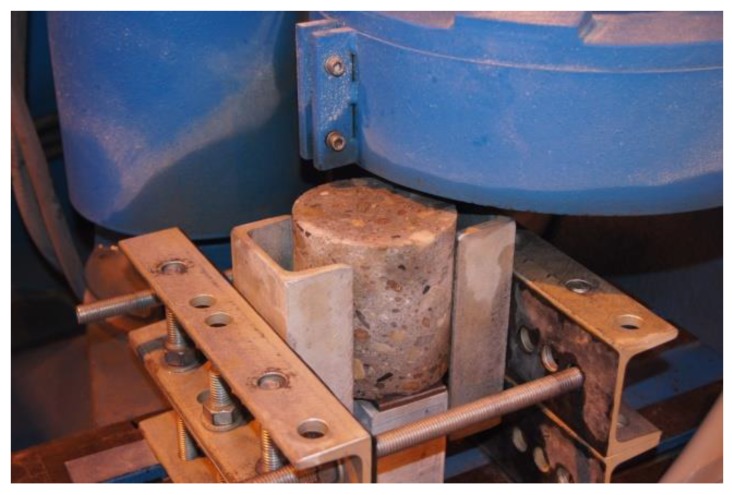
One of the core samples during grinding of its end surface.

**Figure 9 materials-12-02162-f009:**
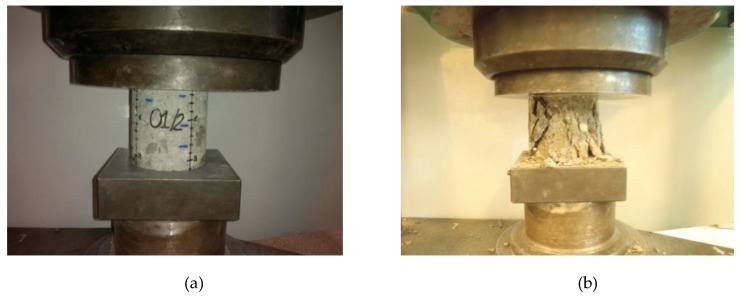
The sample in strength testing machine: (**a**) during loading, (**b**) after failure.

**Figure 10 materials-12-02162-f010:**
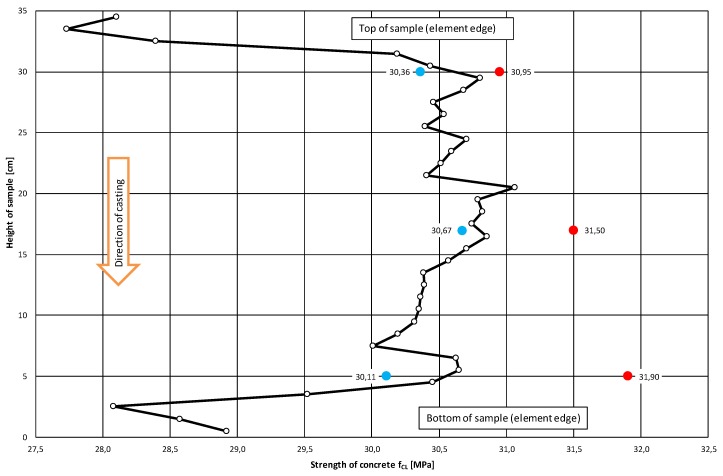
The mean distribution of compressive strength f_CL_ of concrete along the height of the sample taken perpendicularly to top surface of slab. The diagram includes the mean concrete strengths for the top, middle and bottom samples, obtained from ultrasonic tests (marked blue) and destructive tests (marked red).

**Figure 11 materials-12-02162-f011:**
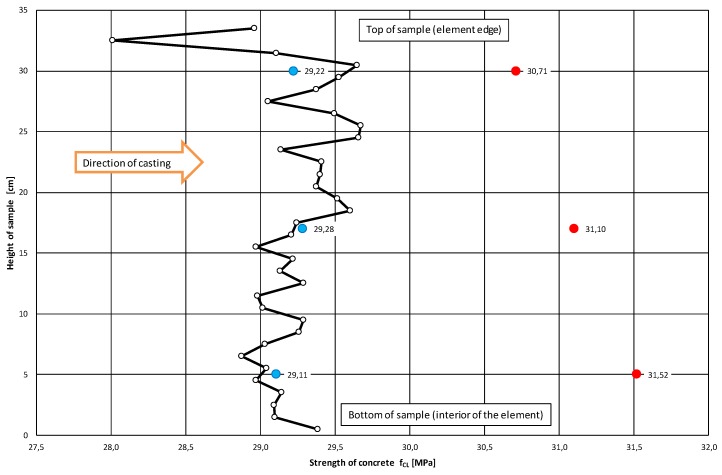
The mean distribution of compressive strength f_CL_ of concrete along the height of the sample taken parallel with the top surface of the slab. The diagram includes the mean concrete strengths for the top, middle and bottom samples obtained from ultrasonic tests (marked blue) and destructive tests (marked red).

**Table 1 materials-12-02162-t001:** The mean dimensions of core samples and their weight.

Sample Number	d_śr_	h_śr_	m	V	ρ
	mm		g	cm^3^	g/cm^3^
01	98.7	351.7	6158.5	2691	2.29
02	98.5	351.4	6167.5	2680	2.30
03	98.6	351.5	6138.0	2682	2.29
04	98.6	351.4	6176.5	2682	2.30
05	98.5	350.0	6121.5	2669	2.29
06	98.6	350.2	6121.0	2675	2.29
07	98.6	350.0	6122.0	2672	2.29
08	98.7	351.3	6172.5	2688	2.30
09	98.5	351.4	6168.2	2678	2.30
10	98.5	350.0	6122.4	2667	2.30
11	98.6	351.5	6140.0	2684	2.29
12	98.5	351.4	6137.6	2678	2.29

**Table 2 materials-12-02162-t002:** Velocities C_L_ ultrasonic wave propagation, measured in two perpendicular directions in core samples taken perpendicularly to top surface of slab.

Measuring Place Distance from Sample Top	Sample 01		Sample 02		Sample 03		Sample 04		Sample 05		Sample 06	
	C_L 1_	C_L 2_	C_L 1_	C_L 2_	C_L 1_	C_L 2_	C_L 1_	C_L 2_	C_L 1_	C_L 2_	C_L 1_	C_L 2_
**cm**	**km/s**											
0.5	3.19	3.39	3.49	3.18	3.32	3.44	3.40	3.31	3.36	3.36	3.23	3.29
1.5	3.18	3.35	3.48	3.02	3.28	3.38	3.29	3.26	3.35	3.30	3.22	3.33
2.5	3.30	3.41	3.49	3.48	3.31	3.37	3.40	3.36	3.28	3.33	3.27	3.36
3.5	3.58	3.63	3.62	3.57	3.43	3.54	3.57	3.62	3.33	3.54	3.55	3.65
4.5	3.60	3.68	3.60	3.61	3.49	3.52	3.53	3.62	3.53	3.52	3.61	3.63
5.5	3.51	3.66	3.64	3.54	3.66	3.62	3.66	3.62	3.62	3.67	3.57	3.61
6.5	3.60	3.68	3.70	3.52	3.57	3.67	3.68	3.44	3.47	3.66	3.60	3.63
7.5	3.53	3.59	3.62	3.48	3.64	3.58	3.58	3.60	3.53	3.62	3.62	3.58
8.5	3.46	3.60	3.72	3.61	3.57	3.50	3.70	3.60	3.54	3.61	3.56	3.58
9.5	3.53	3.57	3.74	3.57	3.26	3.62	3.71	3.58	3.44	3.59	3.60	3.65
10.5	3.61	3.59	3.70	3.57	3.57	3.63	3.54	3.53	3.47	3.58	3.67	3.79
11.5	3.58	3.59	3.62	3.66	3.60	3.62	3.62	3.56	3.50	3.58	3.58	3.62
12.5	3.56	3.65	3.62	3.65	3.55	3.65	3.60	3.58	3.52	3.62	3.41	3.62
13.5	3.32	3.63	3.61	3.60	3.57	3.49	3.65	3.61	3.57	3.57	3.57	3.70
14.5	3.38	3.65	3.72	3.66	3.55	3.63	3.70	3.70	3.63	3.62	3.66	3.76
15.5	3.45	3.67	3.73	3.57	3.65	3.63	3.44	3.53	3.61	3.65	3.73	3.68
16.5	3.60	3.66	3.73	3.58	3.62	3.51	3.72	3.59	3.50	3.61	3.62	3.65
17.5	3.53	3.68	3.72	3.53	3.53	3.62	3.66	3.53	3.54	3.61	3.65	3.70
18.5	3.68	3.67	3.64	3.49	3.62	3.53	3.70	3.59	3.54	3.66	3.61	3.70
19.5	3.58	3.72	3.66	3.59	3.47	3.48	3.65	3.57	3.62	3.66	3.63	3.62
20.5	3.59	3.68	3.63	3.47	3.57	3.39	3.64	3.70	3.43	3.70	3.62	3.66
21.5	3.62	3.57	3.57	3.58	3.53	3.56	3.62	3.54	3.54	3.62	3.48	3.65
22.5	3.40	3.61	3.62	3.61	3.55	3.45	3.70	3.50	3.54	3.61	3.58	3.70
23.5	3.45	3.64	3.62	3.65	3.68	3.59	3.59	3.61	3.40	3.37	3.58	3.65
24.5	3.35	3.62	3.51	3.62	3.52	3.58	3.64	3.70	3.48	3.53	3.62	3.65
25.5	3.55	3.66	3.65	3.48	3.53	3.50	3.53	3.65	3.46	3.57	3.64	3.57
26.5	3.38	3.71	3.69	3.61	3.57	3.32	3.48	3.65	3.44	3.61	3.58	3.58
27.5	3.54	3.55	3.62	3.55	3.54	3.43	3.58	3.63	3.37	3.58	3.54	3.49
28.5	3.54	3.61	3.52	3.57	3.56	3.59	3.62	3.66	3.57	3.66	3.62	3.65
29.5	3.61	3.65	3.63	3.58	3.60	3.59	3.68	3.69	3.48	3.54	3.57	3.57
30.5	3.57	3.62	3.64	3.57	3.57	3.57	3.55	3.62	3.57	3.53	3.61	3.54
31.5	3.45	3.61	3.58	3.52	3.45	3.48	3.57	3.29	3.57	3.50	3.27	3.51
32.5	3.24	3.20	3.52	3.25	3.13	3.34	3.30	3.42	3.48	3.26	3.43	3.34
33.5	3.33	3.25	3.57	3.22	3.28	3.20	3.51	3.42	3.54	3.41	3.42	3.42
34.5	3.39	3.29	3.59	3.23	3.47	3.04	3.62	3.49	3.49	3.49	3.40	3.49

**Table 3 materials-12-02162-t003:** Velocities C_L_ of longitudinal ultrasonic wave propagation, measured in two perpendicular directions in core samples taken parallel with the top surface of the slab.

Measuring Place Distance from Sample Top	Sample 07		Sample 08		Sample 09		Sample 10		Sample 11		Sample 12	
	C_L 1_	C_L 2_	C_L 1_	C_L 2_	C_L 1_	C_L 2_	C_L 1_	C_L 2_	C_L 1_	C_L 2_	C_L 1_	C_L 2_
**cm**	**km/s**											
0.5	3.31	3.42	3.60	3.40	3.34	3.60	3.60	3.35	3.30	3.28	3.50	3.38
1.5	3.27	3.33	3.50	3.28	3.25	3.30	3.32	3.34	3.29	3.26	3.40	3.31
2.5	3.55	3.47	3.31	3.60	3.36	3.56	3.41	3.51	3.32	3.41	3.45	3.33
3.5	3.56	3.45	3.25	3.46	3.59	3.42	3.62	3.53	3.52	3.54	3.43	3.59
4.5	3.54	3.49	3.54	3.42	3.45	3.55	3.50	3.51	3.40	3.41	3.46	3.56
5.5	3.55	3.46	3.46	3.60	3.40	3.50	3.48	3.44	3.41	3.55	3.42	3.36
6.5	3.57	3.50	3.41	3.45	3.51	3.44	3.41	3.44	3.28	3.38	3.46	3.37
7.5	3.49	3.50	3.57	3.42	3.51	3.53	3.44	3.45	3.39	3.50	3.56	3.43
8.5	3.45	3.43	3.56	3.60	3.47	3.54	3.55	3.41	3.49	3.51	3.51	3.49
9.5	3.55	3.43	3.60	3.52	3.44	3.57	3.48	3.51	3.50	3.44	3.45	3.50
10.5	3.40	3.47	3.49	3.50	3.45	3.50	3.40	3.45	3.41	3.49	3.39	3.39
11.5	3.56	3.43	3.46	3.51	3.54	3.46	3.41	3.55	3.36	3.46	3.44	3.50
12.5	3.55	3.38	3.42	3.50	3.54	3.46	3.42	3.47	3.52	3.55	3.43	3.43
13.5	3.57	3.43	3.38	3.46	3.51	3.56	3.37	3.40	3.39	3.50	3.55	3.51
14.5	3.48	3.47	3.41	3.55	3.57	3.51	3.38	3.40	3.51	3.55	3.46	3.52
15.5	3.59	3.47	3.55	3.51	3.43	3.59	3.42	3.41	3.48	3.58	3.44	3.45
16.5	3.54	3.45	3.56	3.46	3.42	3.46	3.38	3.38	3.41	3.45	3.56	3.40
17.5	3.50	3.40	3.53	3.35	3.57	3.56	3.49	3.44	3.40	3.38	3.41	3.39
18.5	3.47	3.44	3.46	3.47	3.49	3.45	3.40	3.36	3.42	3.39	3.38	3.40
19.5	3.42	3.43	3.41	3.55	3.45	3.43	3.40	3.46	3.42	3.50	3.43	3.54
20.5	3.40	3.48	3.47	3.46	3.44	3.50	3.44	3.43	3.38	3.40	3.38	3.55
21.5	3.45	3.50	3.50	3.39	3.42	3.42	3.44	3.56	3.48	3.44	3.44	3.49
22.5	3.40	3.41	3.41	3.42	3.46	3.42	3.40	3.44	3.50	3.43	3.42	3.43
23.5	3.43	3.43	3.41	3.45	3.54	3.41	3.41	3.49	3.33	3.50	3.37	3.41
24.5	3.50	3.38	3.53	3.50	3.38	3.42	3.49	3.42	3.52	3.38	3.51	3.50
25.5	3.42	3.50	3.51	3.38	3.40	3.57	3.46	3.40	3.46	3.41	3.45	3.53
26.5	3.56	3.38	3.45	3.36	3.44	3.37	3.45	3.25	3.55	3.45	3.49	3.44
27.5	3.54	3.45	3.51	3.47	3.39	3.31	3.30	3.36	3.36	3.43	3.41	3.46
28.5	3.42	3.37	3.55	3.46	3.40	3.38	3.48	3.36	3.37	3.37	3.50	3.55
29.5	3.50	3.43	3.46	3.34	3.43	3.36	3.44	3.35	3.43	3.35	3.53	3.50
30.5	3.56	3.50	3.42	3.43	3.45	3.45	3.44	3.53	3.50	3.37	3.37	3.32
31.5	3.57	3.40	3.55	3.33	3.44	3.43	3.46	3.50	3.40	3.37	3.37	3.45
32.5	3.40	3.34	3.40	3.53	3.40	3.41	3.50	3.56	3.40	3.45	3.45	3.44
33.5	3.50	3.40	3.59	3.51	3.47	3.45	3.42	3.49	3.51	3.40	3.44	3.47

**Table 4 materials-12-02162-t004:** The results of concrete compressive strength tests.

Samp. No.	d_m_	h_m_	m	A_c_	V_c_	ρ	F_is_	f_is_	Samp. No.	d_m_	h_m_	m	A_c_	V_c_	ρ	F_is_	f_is_
	mm		g	cm^2^	cm^3^	g/cm^3^	kN	MPa		mm		g	cm^2^	cm^3^	g/cm^3^	kN	MPa
01/1	98.7	99.9	1725	76.51	764	2.26	236	30.85	07/1	98.6	100.0	1760	76.36	764	2.30	226	29.60
01/2	98.7	99.8	1743	76.51	764	2.28	240	31.37	07/2	98.6	99.9	1751	76.36	763	2.30	231	30.25
01/3	98.7	99.8	1765	76.51	764	2.31	242	31.63	07/3	98.6	99.8	1746	76.36	762	2.29	232	30.38
02/1	98.5	100.0	1724	76.20	762	2.26	236	30.97	08/1	98.7	99.9	1755	76.51	764	2.30	230	30.06
02/2	98.5	99.8	1761	76.20	760	2.32	238	31.23	08/2	98.7	99.9	1761	76.51	764	2.30	235	30.71
02/3	98.5	99.9	1762	76.20	761	2.31	243	31.89	08/3	98.7	99.9	1757	76.51	764	2.30	237	30.98
03/1	98.5	99.8	1711	76.20	760	2.25	233	30.58	09/1	98.5	99.7	1744	76.20	760	2.30	234	30.71
03/2	98.5	99.9	1741	76.20	761	2.29	241	31.63	09/2	98.5	99.9	1740	76.20	761	2.29	238	31.23
03/3	98.6	99.6	1768	76.36	761	2.32	248	32.48	09/3	98.5	100.0	1760	76.20	762	2.31	243	31.89
04/1	98.5	99.9	1731	76.20	761	2.27	237	31.10	10/1	98.5	99.6	1754	76.20	759	2.31	239	31.36
04/2	98.5	99.9	1755	76.20	761	2.31	240	31.50	10/2	98.5	99.8	1742	76.20	760	2.29	245	32.15
04/3	98.5	99.8	1779	76.20	760	2.34	240	31.50	10/3	98.5	99.8	1748	76.20	760	2.30	248	32.55
05/1	98.5	99.9	1717	76.20	761	2.26	234	30.71	11/1	98.6	99.7	1741	76.36	761	2.29	237	31.04
05/2	98.5	99.9	1745	76.20	761	2.29	240	31.50	11/2	98.6	99.7	1747	76.36	761	2.29	232	30.38
05/3	98.5	99.9	1769	76.20	761	2.32	242	31.76	11/3	98.6	99.9	1750	76.36	763	2.29	238	31.17
06/1	98.5	99.9	1716	76.20	761	2.25	240	31.50	12/1	98.5	99.9	1766	76.20	761	2.32	240	31.50
06/2	98.5	99.9	1741	76.20	761	2.29	242	31.76	12/2	98.5	99.8	1754	76.20	760	2.31	243	31.89
06/3	98.5	99.9	1762	76.20	761	2.31	245	32.15	12/3	98.5	100.1	1750	76.20	763	2.29	245	32.15

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
