# Peer review of "Variation in Compressive Strength of Concrete aross Thickness of Placed Layer"

_materials, 2019, doi:10.3390/ma12132162_

Reviewer 1 Report

the authors report on an interesting topic in a well-organised way. The content is clear and the results consistent. in this sense the paper can be accepted in the present form. however, the support to the conclusion appear very poor since only sonic tests were performed. in order to make the investigation sufficently appreciable, other different test needs to be perfomed and compared with the sonic in order to look for convergence of findings.

Author Response

Thank you very much for your valuable reviewer's comments. Indeed, performing other tests could convince about the correctness of ultrasonic measurements. At the moment (after completing non-destructive and destructive testing), performing other tests is difficult. However, this may be the subject of further research.

To confirm the obtained values (tables 2 and 3) of the CL velocity of the longitudinal wave propagation, a standard model was used (for low vibration frequencies), where the velocity of propagation of different waves in the medium can be determined from the formula CL = (Ecm / r)1/2. In this formula, Ecm is the average static modulus of elasticity of concrete, and r the density of concrete. For the experimentally determined (on cubic samples formed during the production of the concrete slab) class C25/30 concrete, the standard elastic modulus is Ecm = 31 GPa, and the concrete density r= 2,29 g/cm3 is given in table 1.  Therefore the CL velocity of the longitudinal wave propagation is CL = (31´109 / 2290)1/2 = 3679 m/s = 3,68 km/s and is similar to the speed obtained from ultrasonic measurements (tables 2 and 3).

 I hope that the above explanations will be sufficient

 Jarosław Michałek

Reviewer 2 Report

In this paper, the strength distribution into the slab depth direction is analyzed by ultrasonic velocity test and compressive strength test on the drilled cores. Please correct or reply to the followings.

1. Please, provide the information of the mix design.

2. Please, add the details of the ultrasonic test, such as input voltage, PRF, the energy attenuation , etc.

3. How can you obtain the consistency of the measured values during the ultrasonic test?

4. When the compressive strength is about 30MPa, the P wave velocity is more than 4000m/s. The P wave velocities in Table 2 and Table 3 seem to be relatively small.

5. 01 ÷ 06, 07 ÷ 12 in the manuscript?

6. It is possible to predict the compressive strength according to ASTM C 39 by taking 100 mm x 200 mm specimens or by cutting 100 mm x 200 mm of the drilled specimens. Why did you estimate the compressive strength through complicated correction for 100 mm x 100 mm?

7. Dynamic modulus measurement is possible by ASTM C 215 on the specimens, and P-wave velocity can be calculated by the estimated modulus. It is necessary to verify the value of the measured P wave velocity by ASTM C 215 or other methods.

8. Overall, it seems to be difficult to detect a difference of 1~2 MPa by the P-wave velocity. Please answer this.

Author Response

Thank you very much for your valuable reviewer's comments. As far as possible, I present the answers to each of the comments with the hope that they will be sufficient.

1) A concrete with the following composition per 1 m3 was used to make the slab from which the test samples were taken:

-        cement CEMII / BS 32.5 - 270 kg,

-        addition in the form of fly ash - 60 kg,

-        plasticizer - 2.43 kg,

-        aggregate (sand 0/2 mm - 40%, gravel 2/8 mm - 26%, gravel 8/16 mm - 34%) - 1879 kg,

- water - 170 kg.

2) Details of the use of the Unipan Materials Tester Type 543 with exponential warheads with point contact with the surface to be tested and the frequency of 40 kHz are described in the paper [15] quoted below: Gudra T., Stawiski B .: Non-destructive strength characterization of concrete using Surface waves, NDT & E International 33 (2000), pp. 1-6. The article contains only the most important information on the ultrasonic tests.

3) The CL velocities of longitudinal wave propagation obtained from ultrasonic testing for samples (01 ÷ 06) taken perpendicular to the upper surface of the plate and samples (07-12) taken in parallel to the upper surface of the panel are very repeatable (excluding the disturbance zones). This is evidenced by low values of standard deviation sCL and coefficient of variation nCL. The obtained values of the average velocity CL of ultrasonic wave propagation and standard deviations sCL as well as variability coefficients nCL are given in the article text (lines 129 ÷ 135)

4) One can actually find in the literature (eg Neville A.M. Properties of Concrete, Polski Cement Sp. z oo, Cracow, 2000) relationships between the concrete compressive strength and the velocity of ultrasonic wave propagation indicating that for CL = 3500 ÷ 4000 m/s a concrete with compressive strength of, roughly, up to 25 MPa should be obtained. In this case, for average approximate speeds CL = 3500 m/s, the average concrete strengths fm,is = 31 MPa were obtained. It should be noted, however, that there is no single and detailed relationship between the velocity of propagation of the ultrasonic wave and the compressive strength of concrete, and the descriptions used are highly approximate.

5) Samples with numbers (01 ÷ 06) were taken perpendicular to the top surface of the slab, and samples with numbers (07 ÷ 12) parallel to the top surface of the slab.

6) Core samples with a diameter d = 100 mm were taken from a slab of thickness h = 350 mm (Table 1). For the destructive tests, these samples were cut into smaller ones (Fig. 7) with the height h = 100 mm (h/d = 1). In the outer samples (e.g. 01/1 - top sample, 01/3 - bottom sample) the original external surfaces were left (only sometimes slightly cut for levelling). The middle sample (e.g. 01/2) was cut to the required size of 100 mm. In accordance with the PN-EN 12504-1:2001 Testing concrete in structures. Part 1: Cored specimens. Taking, examining and testing in compression, in the case of length to diameter ratio of the core equal to 1 (as in this case), the result of the measurement of the concrete compressive strength will refer to the strength tested on cubic samples. Additionally, the PN-EN 13791:2008 Assessment of in-situ compressive strength in structures and precast concrete components states that when the compression strength of concrete in a structure is determined on cored boreholes with the length equal to the diameter 100 mm, the value of strength which is obtained, corresponds to the strength of a cubic sample with a side equal to 150 mm.

7) The CL velocity value of the longitudinal wave propagation was not verified using the resonance frequency method to determine the dynamic modulus of elasticity of concrete. This may be the subject of further tests. To confirm the obtained (Tables 2 and 3) speeds CL of longitudinal wave propagation, a standard model (for low vibration frequencies) was used, where it is possible to determine, as in the case of a linearly elastic body, the velocity of propagation of various waves in the medium from the formula CL = (Ecm / r)1/2. In this formula, Ecm is the mean static modulus of elasticity of concrete, and the r is density of concrete. For the experimentally determined (on cubic samples formed during the production of the concrete slab) class C25/30 concrete, the standard elastic modulus is Ecm = 31 GPa, and the concrete density r= 2,29 g/cm3 is given in table 1. Therefore, the speed CL of longitudinal wave propagation is CL = (31´109 / 2290)1/2 = 3679 m/s = 3,68 km/s and is similar to the speed obtained from ultrasonic measurements (Tables 2 and 3).

8) When analyzing the results of ultrasonic tests it should be remembered that this is an approximate method of assessing the compressive strength of concrete with many limitations. The change of strength within the limits of 1-2 MPa is irrelevant from the point of view of the accuracy of the testing method.

I hope that the above explanations will be sufficient and that the additions introduced to the text will significantly improve the clarity and legibility of the paper.

Regards

Jarosław Michałek

Round  2

Reviewer 1 Report

The answer of the author  is satisfying